# Plant DNA Repair and *Agrobacterium* T−DNA Integration

**DOI:** 10.3390/ijms22168458

**Published:** 2021-08-06

**Authors:** Stanton B. Gelvin

**Affiliations:** Department of Biological Sciences, Purdue University, West Lafayette, IN 47907-1392, USA; gelvin@purdue.edu

**Keywords:** *Agrobacterium*, chromatin, DNA polymerase θ, DNA repair, genome rearrangements, microhomology−mediated end−joining (MMEJ), non-homologous end-joining (NHEJ), T−DNA borders, T−DNA integration

## Abstract

*Agrobacterium* species transfer DNA (T−DNA) to plant cells where it may integrate into plant chromosomes. The process of integration is thought to involve invasion and ligation of T-DNA, or its copying, into nicks or breaks in the host genome. Integrated T−DNA often contains, at its junctions with plant DNA, deletions of T−DNA or plant DNA, filler DNA, and/or microhomology between T-DNA and plant DNA pre-integration sites. T−DNA integration is also often associated with major plant genome rearrangements, including inversions and translocations. These characteristics are similar to those often found after repair of DNA breaks, and thus DNA repair mechanisms have frequently been invoked to explain the mechanism of T−DNA integration. However, the involvement of specific plant DNA repair proteins and *Agrobacterium* proteins in integration remains controversial, with numerous contradictory results reported in the literature. In this review I discuss this literature and comment on many of these studies. I conclude that either multiple known DNA repair pathways can be used for integration, or that some yet unknown pathway must exist to facilitate T−DNA integration into the plant genome.

## 1. Introduction

The process of *Agrobacterium* T-DNA integration into the genomes of infected plants has important implications both for understanding plant DNA break repair processes and for the use of *Agrobacterium* as a tool for manipulating the plant genome. Although scientists have studied T-DNA integration for decades, we still have a very incomplete picture of how integration occurs, and numerous contradictory conclusions abound in the literature. In this short article, I summarize and analyze some of these conclusions, and relate what we know about plant DNA repair processes to possible mechanisms of T-DNA integration. For past discussions regarding the mechanism of T-DNA integration, I refer the reader to [1,2,3,4,5,6,7,8,9,10].

## 2. Are *Agrobacterium* Proteins Involved in T-DNA Integration into the Plant Genome?

The pathway of T-DNA transfer from *Agrobacterium* to plant cells, and its ultimate integration into the plant genome, starts with nicking the T-DNA region of a Ti (tumor inducing) or Ri (rhizogenic) plasmid by the T-DNA border-specific endonuclease VirD2 [11,12,13,14,15]. Nicking occurs between nucleotides 3 and 4 of the 25 bp border sequences that flank the T-DNA region [16,17]. During T-DNA border nicking, VirD2 covalently links to the 5′ end of T-DNA, resulting in a single-strand form of T-DNA, the T-strand, that on its 3′ end contains nucleotides 4–25 of the left border (LB), and on its 5′ end contains nucleotides 1–3 of the right border (RB) [18,19,20,21,22]. VirD2 subsequently leads the T-strand through a dedicated type IV protein secretion system (T4SS) and into the plant cell [23,24]. Within the plant, T-strands may suffer deletions at the 3′ and/or 5′ ends before or during integration. Deletions are especially prevalent, and generally more extensive, at the 3′ end than at the 5′ end, which is protected by its linkage to VirD2 protein (e.g., [21]).

Does VirD2 directly participate in the T-DNA integration process? VirD2 can both cleave and re-ligate (i.e., reverse the reaction) T-DNA border sequences in vitro [25]. However, VirD2 does not harbor an activity that can ligate VirD2/T-strands to generalized target sequences; this could only be done by a ligase activity found in plant extracts [26].

VirD2 contains a highly conserved region (amino acids DGRGG) near the C-terminus, termed the ω domain [27]. Substitution of DDGR (the first D is not part of ω) by four serine residues resulted in a mutant VirD2 protein that conferred somewhat lower transient transformation activity (~20–30% of wild-type levels) upon its *Agrobacterium* host compared with wild-type VirD2. However, stable transformation was reduced by >95% [27,28,29]. Although stable transformation of plants using this VirD2 ω mutant was decreased, the precision of the integration of sequences near the RB was similar to that observed when using wild-type VirD2 [29]. Mutation of other sequences in VirD2 protein, however, could alter the precision of T-DNA integrated near the RB [30]. Taken together, the results of these two studies suggest that VirD2 may be involved in T-DNA integration.

Although the VirD2 ω mutant *Agrobacterium* strain identified by Shurvinton et al. [27] showed moderately lower transient transformation activity but greatly reduced stable transformation, two other ω domain mutants (a precise deletion of the DGRGG ω amino acids, or their replacement with five glycine residues) resulted in both greatly decreased transient and stable transformation frequency [31]. The different relative transformation activities conferred by the various VirD2 ω mutants may result from altered protein structure conferred by the serine residue substitutions.

## 3. Where in the Plant Genome Does T-DNA Integrate?

Early studies indicated that T-DNA integration is random at the chromosome level [32,33,34]. Generation of numerous *Arabidopsis* and rice T-DNA insertion libraries, each consisting of tens of thousands of individually tagged plant genomes, allowed the first large scale probing of T-DNA insertion locations at the DNA sequence level. Results of these studies initially indicated that T-DNA preferentially integrated into transcriptionally active genes, promoter regions, or sequences of high A+T content [35,36,37,38,39,40]. These studies, however, all suffered from the problem of selection bias; the individual transgenic plants each harboring a different T-DNA integration event had been selected for antibiotic/herbicide resistance. If T-DNA had integrated into a transcriptionally inert region of the plant genome, the selection marker gene would not have been expressed and the resulting transgenic plant would have been lost. Indeed, fewer T-DNA insertions into heterochromatic regions of DNA, centromeres, telomeres, and rRNA genes were recovered relative to the proportion of the genomes represented by these sequences, resulting in the appearance of T-DNA integration into only transcriptionally active regions of the genome [37].

In contrast to the studies cited above, two groups examined T-DNA integration sites in the *Arabidopsis* genome in which cells were not selected for expression of any transgene, including those for antibiotic/herbicide resistance [41,42]. These experiments indicated that T-DNA did not preferentially integrate into any particular sequence context or region of gene expression. For example, approximately 10% of the *Arabidopsis* genome is composed of highly repeated DNA sequences, and ~10% of the insertions occurred in these sequences. T-DNA insertions into rDNA, centromeres, and telomeres also approximated their relative proportion of the genome. T-DNA pre-integration sites were average in their extent of transcription and methylation, as compared with the entire genome [41]. Thus, T-DNA integration did not preferentially occur into any particular chromosome sequence or feature. These results were reproduced using a high-throughput sequencing strategy to identify, without selection, T-DNA integrated into the *Arabidopsis* genome within six hours after infection [42]. This group also failed to identify preferential T-DNA integration into particular sequences, and the extent of DNA methylation of pre-integration sites was not biased. They did identify a slight local A+T motif enrichment at the pre-integration site, and microhomology was often observed between the T-DNA border sequences and the pre-integration site. Microhomology between T-DNA border regions and pre-integration sites is a common feature of T-DNA integration (e.g., [43,44,45]). The one distinct feature of pre-integration sites was a high nucleosome occupancy and a high level of histone H2K27 trimethylation. However, data for these last two items were based on database entries and not on direct analysis of chromatin from the tissues the authors used in their studies.

Taken together, these two studies do not point to any distinctive plant DNA or chromatin features as T-DNA target integration sites. However, chromatin conformation may influence T-DNA integration. The *rat5 Arabidopsis* mutant is highly susceptible to transient transformation but highly resistant to stable transformation by *Agrobacterium* [46]. This mutant contains a T-DNA insertion in the 3′ untranslated region of the histone H2A-1 gene *HTA1* [47]. Overexpression of *HTA1* in otherwise wild-type *Arabidopsis* plants increased stable transformation, as it also did in rice [48]. The expression of *HTA1* correlates with *Arabidopsis* cell and tissue susceptibility to *Agrobacterium*-mediated transformation [49], and all tested histone H2A proteins were functionally redundant with respect to increasing transformation when expressed in *Arabidopsis* [50]. In addition to histone H2A, overexpression of histone H4 and one histone H3 protein (HTR11) also conferred hyper-susceptibility to *Agrobacterium*-mediated transformation. However, overexpression of other histone H3 proteins and all tested histone H2B genes had no effect on transformation [51]. It is not clear whether manipulation of histone levels in plant cells directly alters the extent of T-DNA integration, or whether these altered histone levels influence plant gene expression, thereby resulting in T-DNA integration [52]. Nevertheless, the direct interaction of VirD2 with histones in yeast suggests that histones may help direct VirD2/T-strand complexes to the host genome [53].

Do *Agrobacterium* or host proteins guide T-strands to host chromatin prior to integration? VirE2 protein, a virulence effector protein secreted by *Agrobacterium* into plant cells, is a single-strand DNA-binding protein that has been proposed to interact with VirD2/T- strands in the plant cell, forming a “T-complex” [54]. (It should be noted that such complexes have never been identified in *Agrobacterium*-infected plants). VirE2 can interact with the plant bZIP transcription factor VIP1 [55]. VIP1 can interact with histones and nucleosomes, suggesting that VIP1 may be a molecular link between T-complexes entering the nucleus and T-DNA integration sites in the plant chromosomes [56,57,58]. However, two recent publications demonstrated that neither VIP1 nor its orthologs are required for *Agrobacterium*-mediated transformation, throwing into doubt a role for VIP1 in T-DNA integration [59,60].

In addition to the involvement of histones in *Agrobacterium*-mediated transformation and, perhaps, T-DNA integration, histone-associated and -modifying proteins also affect transformation. Crane and Gelvin [61] tested RNAi lines individually directed against 109 *Arabidopsis* chromatin-related genes for transient and stable transformation. Silencing of 24 of these genes decreased transformation. In particular, silencing of *SGA1* (encoding a histone H3 chaperone) and *HDT1* and *HDT2* (encoding histone deacetylases) greatly decreased both stable transformation and T-DNA integration. Silencing of genes involved in chromatin remodeling, DNA methylation, histone acetylation, and nucleosome assembly also had an effect on stable transformation, although this may result from secondary effects these genes have on the expression of other genes involved in stable transformation. Deletion of genes encoding components of yeast histone acetyltransferase complexes increased yeast transformation, whereas deletion of genes encoding proteins of histone deacetylase complexes decreased yeast transformation [62]. For some of these mutants, integration of T-DNA into the yeast genome was disrupted.

Taken together, these results indicate that proteins associated with chromatin structure and modification are important for T-DNA integration into plant or yeast genomes, and that in some instances, the effect on integration may be direct rather than indirectly influencing the expression of other genes important for T-DNA integration.

As described above, the position of T-DNA integration, and the chromatin structure of that region, may influence the expression of T-DNA-encoded transgenes. As a practical consideration, this variability in transgene expression (the so-called “position effect”) will influence studies on gene and promoter function. Scientists therefore need to examine a large number of independent transgenic events to draw conclusions about, e.g., relative promoter strengths.

## 4. Give Me a Break (?)

Integration of naked DNA can be increased by low-dose X-irradiation of transfected plant protoplasts, suggesting that generation of DNA breaks enhances foreign DNA integration [63]. Such integration events frequently result in deletions and rearrangements at the integration site. Furthermore, filler DNA from either within the introduced DNA or from elsewhere in the plant genome often appears at plasmid–plasmid junction sites, as does the presence of microhomology between molecules at the junction site [64]. Such genomic rearrangements are reminiscent of those caused by radiation-induced damage in *Arabidopsis* [65], suggesting that naked DNA integration into the plant genome occurs at DNA double−strand break sites. Similar rearrangements are frequently detected at junctions between integrated T-DNA and plant DNA, and between T-DNA borders of T-circles (see below). Thus, integration of various forms of exogenous DNA likely occurs by similar mechanisms, regardless of whether the introduced DNA is in a “naked” double-strand form or in the form of single-strand T-complexes after transfer from *Agrobacterium*. DNA break repair models have been invoked to understand how each of these forms of DNA integrates into the plant genome [66].

DNA repair and recombination processes require breaks in the phosphodiester DNA backbone, either in the form of single-strand nicks or double-strand breaks. Numerous models for T-DNA integration have been proposed; all these models incorporate, as part of their mechanism, nicks or breaks in the host target DNA at the site of T-DNA integration (see, e.g., [1,3,4,5,6,7,8,10]).

T-DNA preferentially integrates into double-strand DNA breaks [67]. This observation was followed by two other reports also showing preferential T-DNA integration into double-strand break sites [68,69]. In each of these studies, a rare cutting meganuclease (either I-*Sce*I or I-*Ceu*I) was used to cut tobacco DNA during transformation. T-DNA was preferentially “trapped” in these cut sites at frequencies up to several percent of the examined integration events. More recently, scientists used CRISPR technology to generate double-strand breaks in DNA, either to generate site-directed mutations or to attempt homology-dependent repair using recombination with correction templates. In several instances, T-DNA was trapped at these break sites following Cas nuclease cutting (e.g., [70,71]). It is thus clear that double-strand DNA breaks can act as a “T-DNA magnet”. However, does *Agrobacterium* take advantage of naturally occurring host DNA breaks (or nicks), or can *Agrobacterium* infection perhaps induce host DNA disruptions?

That *Agrobacterium* can incite DNA breaks would not be unusual, because inoculation by other plant pathogens (bacteria, oomycetes, and fungi) can cause double-strand DNA breaks in host plant genomes [72]. DNA disruptions occur in *Arabidopsis* cells near the site of *Agrobacterium* infection, as detected by COMET assays. However, because alkaline pH conditions were used in this study, it is not clear whether these disruptions resulted from single-strand nicks or double-strand breaks in the plant DNA [73]. Recent results indicate that *Arabidopsis* cells, exposed to *Agrobacterium* but not stably transformed, contain a higher number of in/dels than would be expected from the natural frequency of such mutations [74]. These results suggest that incubation of cells with *Agrobacterium* is inherently mutagenic, causing double-strand DNA breaks that are mis-repaired.

There are many hints in the literature that *Agrobacterium* infection can cause mutations independent of T-DNA integration; these mutations may result from induced double-strand DNA breaks that are subsequently mis-repaired. They may also be generated by “abortive integration” of T-DNA, followed by mis-repair of the abortive integration site. For example, *N. plumbaginifolia* plants, containing one mutant nitrate reductase (*NR*) gene, could be converted to fully NR null mutants (chlorate resistant) following *Agrobacterium*−mediated transformation. However, none of these null mutants contained T-DNA in the *NR* gene [75]. Mutation of the wild-type *NR* allele must have occurred by some other mechanism.

Another indication that *Agrobacterium* infection may be inherently mutagenic derives from the observation that only ~35% of the T-DNAs in *Arabidopsis* T-DNA insertion libraries co-segregate with a screened mutant phenotype [76,77]. Mutations in the selected lines may be derived from disruptions other than T-DNA insertion into the gene of interest.

Many *Arabidopsis* T-DNA insertion lines contain complex host genome rearrangements that are frequently associated with mis-repair of double-strand DNA breaks. These include inversions, translocations, and other complex rearrangements [78,79,80,81,82,83,84,85]. Similar rearrangements have been detected in transgenic rice [86]. Clark and Krysan [87] noted that approximately 19% of the examined lines from the SALK T-DNA mutant collection contained translocations. The rearrangements of plant genomes following T-DNA integration are reminiscent of the process of chromothripsis resulting from CRISPR−Cas9 mammalian genome editing [88].

## 5. What Is the Mechanism of T-DNA Integration?

Perhaps the most important problem remaining in understanding *Agrobacterium*-mediated transformation is the mechanism of T-DNA integration. As cited above, numerous models of integration have been proposed. What is clear is that homologous recombination is not the mechanism: Despite many kilobases of homology between plant DNA and engineered T-DNA, integration into homologous sequences in the plant genome occurs extremely rarely. This differs from the situation in yeast, where homologous recombination is predominant when homology between T-DNA and the yeast genome is present (see, e.g., [89,90,91]. Thus, what remains for the T-DNA integration mechanism in plants is some form of non-homologous end-joining (NHEJ) in which T-DNA integration occurs in the absence of large regions of homology, although targeting by microhomology may be used in some circumstances.

Two major NHEJ pathways have been described (e.g., [92,93,94,95]). The “classical” (Ku−dependent) pathway utilizes, among other proteins, the Ku70/Ku80 heterodimer to protect the broken DNA ends, and the complex of XRCC4/XLS/DNA ligase IV to repair the break. It is not unusual that, following repair, small deletions, insertions, or nucleotide substitutions occur at or near the break site. Microhomology between the ligated ends is rarely detected. An “alternative” pathway uses microhomology between a region at or near the break site and another sequence (near or distant from the break site) for repair. Participants in this pathway include members of the MRN complex that process broken chromosome ends, the WRN helicase, and a complex of XRCC1 and DNA ligase III (not found in plants) to repair the breaks. DNA polymerase θ is a participant in this pathway and is proposed to play a key role in T-DNA integration. Microhomology-mediated end-joining (MMEJ) is frequently referred to as theta-mediated end-joining because of DNA polymerase θ’s role in this process. DNA polymerase θ has several unusual properties: the protein is made up of both a helicase and a DNA polymerase domain, and the enzyme has a propensity to “template switch”. This latter property allows it to copy DNA from another region of the genome into break sites, generating “filler” DNA sequences in the break. MMEJ is highly mutagenic, frequently generating deletions as sequences flanking DNA break sites search for homologous sequences with which to join. MMEJ also generates chromosomal rearrangements such as inversions and translocations, features commonly associated with T-DNA integration.

Which of these NHEJ pathways, if any, are involved in T-DNA integration? Numerous studies have been published testing stable transformation efficiencies and T-DNA integration characteristics of various *Arabidopsis* and rice NHEJ mutants [6,96,97,98,99,100,101,102,103,104,105,106]. However, with the exception of three publications [102,104,106], all other studies used the frequency of stable transformation as a proxy for T-DNA integration. While detection of stable transformants requires T-DNA integration, it also requires expression of selection marker genes to recover transformed tissue. T-DNA integration may thus occur in the absence of stable transformation if selection marker genes have been silenced. As noted above, such selection bias can confound experimental interpretations [41,42,107]. An additional complication is that most *Arabidopsis* stable transformation experiments were conducted using a flower-dip protocol. It is well-documented that the importance of *Arabidopsis* genes essential for somatic cell transformation differs from that of germ-line transformation [106,108,109]. Finally, stable transformation efficiencies must be calculated with respect to transient transformation frequencies; a decrease in stable transformation may not indicate that a particular plant mutant has altered stable transformation characteristics if the transient transformation frequency is correspondingly altered. It is particularly important that plant inoculations be conducted with several orders of magnitude different *Agrobacterium* concentrations to avoid a “saturation response” with high bacterial inoculum conditions, thus obscuring differences among wild-type and mutant plant genotypes. 

In light of these numerous variables and limitations, it may not be surprising that different laboratories have come to different conclusions with regard to the importance of various plant NHEJ genes for T-DNA integration (or rather, for most studies, stable transformation). Several reports indicate that mutation of the *Arabidopsis* or rice classical (c)NHEJ pathway genes *Ku70*, *Ku80*, or DNA ligase IV (*Lig4*) resulted in lower stable transformation frequencies [6,96,99,100,103]. These studies suggest that these cNHEJ genes are important for T-DNA integration. Other publications indicated that such mutations had little or no effect on stable transformation [97,98]. These studies suggest that these cNHEJ genes are not essential for T-DNA integration. Still other publications, using both *Arabidopsis* and *N. benthamiana*, showed that mutation or down-regulation of several cNHEJ genes, including *Ku70*, *Ku80*, *XRCC4*, and the gene encoding DNA ligase VI (*Lig6*), increased both stable transformation and T-DNA integration into non-selected plant cells [102,104]. These studies suggest that expression of these cNHEJ genes inhibits T-DNA integration, perhaps by speeding the repair of double-strand DNA breaks required for T-DNA integration.

Similarly, individual mutation of two genes associated with MMEJ, *XRCC1* and *PARP2*, did not decrease stable transformation of *Arabidopsis* root tissue ([104]; *PARP1* described in this study has more recently been termed *PARP2*). Mutation of *PARP2* actually increased the frequency of T-DNA integration into the genome of non-selected root cells 2- to 10-fold. The discrepancy between increased T-DNA integration frequency and similar stable transformation frequency of wild-type and *parp2* mutant roots was explained by increased DNA methylation of T-DNA in the *parp2* mutant plants, likely resulting in silencing of the selection genes. This result indicates the importance of investigating T-DNA integration biochemically in non-selected tissue, rather than relying on stable transformation frequency of selected tissue as a proxy for T-DNA integration.

## 6. The Importance of DNA Polymerase θ for *Agrobacterium*-mediated Transformation and T-DNA Integration

In 2016 van Kregten et al. [7] published a seminal paper in which they proposed an essential function for DNA polymerase θ in stable transformation of *Arabidopsis* and T-DNA integration into its genome. These authors examined two DNA polymerase θ (*polQ*) mutants, *tebichi (teb) 2* and *teb5.* Although they could not detect differences in transient transformation between wild-type and *polQ* mutant plants, they were not able to obtain any stable transformants of the *polQ* mutants using either a flower-dip transformation protocol or a root transformation protocol requiring selection of transgenic calli and regeneration of plants from these calli. The authors noted that DNA polymerase θ can “template switch” during DNA replication, and that it can thereby generate “filler” DNA sequences, a common characteristic of T-DNA/plant DNA junctions at the break site, by copying and joining T-strand DNA and microhomologous plant DNA. They also noted that copying T-strand sequences into both ends of a plant DNA double-strand break could result in integration of T-DNA “head-to-head” (RB-to-RB) dimers, also a common characteristic of many T-DNA insertions (Figure 1). T-DNA integration via theta-mediated end-joining thus became the favored model for T-DNA integration into plant genomes.

Nishizawa-Yokoi et al. [106] re-examined the role of DNA polymerase θ in T-DNA integration. Using the same *Arabidopsis teb2* and *teb5* mutants used by van Kregten et al. [7], as well as three independent rice *polQ* mutants, this group was able to obtain stable transformants of somatic tissue in all tested *polQ* mutants. Similar to van Kregten et al. [7], they were not able to transform *Arabidopsis* by a flower-dip protocol, except when the incoming T-DNA constitutively expressed a wild-type *PolQ* gene. These authors additionally showed that transient transformation of roots from the *Arabidopsis polQ* mutants did decrease relative to transformation of wild-type roots. T-DNA/plant DNA junctions isolated from transformed rice and *Arabidopsis polQ* mutant calli had characteristics similar to those isolated from wild-type tissue. Finally, the authors showed that both *Arabidopsis* and rice *polQ* mutants had growth and/or developmental defects; root segments from *Arabidopsis polQ* mutants did not form callus well and the calli grew slowly. Calli derived from rice *polQ* mutants did not regenerate plants even under non-transformation and non-selection conditions. The variable penetrance of the tebichi phenotype was recently examined and was shown to increase under stress, including replication stress, conditions [110,111]. Similar to the situation with *Arabidopsis* flower-dip transformation, rice *polQ* mutants could be transformed and regenerated into plants if the incoming T-DNA contained a constitutively expressed *PolQ* gene (Figure 2). Thus, transformation and developmental deficiencies resulting from mutation of *PolQ* could be complemented by transient expression of a wild-type *PolQ* gene in both *Arabidopsis* and rice.

What were the differences between the experiments conducted by these two groups? For *Arabidopsis* transformation, both groups used the same mutant lines, *teb2* and *teb5*. The major difference was the assays used to determine transient and stable transformation. Van Kregten et al. [7] used a stable transformation assay in which transformed *Arabidopsis* root segments were regenerated to green plants under phosphinothricin selection conditions (the T-DNA they used contained a *bar* gene). Nishizawa-Yokoi et al. [106] used an assay in which root segments were regenerated only to the callus stage. Selection consisted of either crown gall tumor formation on phytohormone-free medium (using an oncogenic *Agrobacterium* strain), or callus induction medium containing phosphinothricin (using an *Agrobacterium* strain with a *bar* gene in the T-DNA region). Tumors and calli from the *teb* mutant roots grew more slowly than did tumors/calli from wild-type plants, reflecting the growth phenotype of *polQ Arabidopsis* mutants. However, stable transformants were detected at the frequency of 25–55% that of wild-type plants. Growth of calli, instead of regeneration of selected plants, thus allowed recovery of transformed tissue for further analysis. The average amount of T-DNA integrated into the genome of *teb* mutant calli, that had not been selected for transformation (to avoid selection bias), was 50–90% that of wild-type calli, as determined by droplet digital PCR. Thus, T-DNA integration into the genome of *Arabidopsis polQ* mutant cells was similar to that of wild-type cells. However, it was more difficult to recover tissue from *polQ* mutants than from wild-type tissue to permit molecular analysis.

A second difference between the groups was the assays for transient transformation. Van Kregten et al. [7] did not see any difference between the wild-type and *teb* mutant plants, whereas Nishizawa-Yokoi et al. [106] saw a decrease in transient transformation of the mutants. Transformation assays can be very sensitive to the bacterial inoculum used; high *Agrobacterium* inoculum concentrations, as used by Van Kregten et al. [7], can mask differences between wild-type and mutant plant responses. Nishizawa-Yokoi et al. [106] used 10-fold serial dilutions of bacteria, starting at a lower bacterial concentration than that used by Van Kregten et al., and were easily able to quantify differences in transient transformation frequencies of wild-type and *teb* mutant plants.

A third difference between the groups was that Nishizawa-Yokoi et al. [106] additionally showed that mutation of *polQ* in rice also allowed stable transformation, although at a reduced frequency to that of wild-type rice tissue.

## 7. T-DNA Integration: An Octopus or a Nopalus?

One could consider at least two explanations for the discrepancies in the literature regarding the importance of individual DNA repair genes and pathways for T-DNA integration: (1) multiple DNA repair pathways could be used for integration. No one pathway is essential, but many could contribute. Thus, like the multiarmed octopus, T-DNA may “grab” any pathway it can find to integrate. This model would predict that disruption of any one pathway, or even simultaneous disruption of multiple pathways, would perhaps decrease but not eliminate T-DNA integration, depending on the importance of that pathway for integration; or (2) none of the known DNA repair pathways play a role in T-DNA integration. Like the mythical nopalus, some as yet unknown pathway mediates integration (Figure 3).

What are the data to suggest that multiple DNA repair pathways may contribute to T-DNA integration? Several groups have reported the effect of simultaneous mutation of multiple DNA repair genes, functioning in different repair pathways, on stable transformation or T-DNA integration. Simultaneous mutation of *Arabidopsis ku80* (important for classical NHEJ) and *parp2* (important for alternative NHEJ) had little effect on either stable transformation of *Arabidopsis* root segments or T-DNA integration into the *Arabidopsis* genome [106]. Mestiri et al. [103] examined transformation of various combinations of *Arabidopsis* DNA repair mutants, including *ku80* (important for the classical NHEJ pathway), *xrccI* (important for the alternative or MMEJ pathway), *xpf* (important for NHEJ and homologous recombination (HR)), and *xrcc2* (important for HR). Increasing the number of mutant repair/recombination genes (from *ku80*, to *ku80*/*xrccI*, to *ku80*/*xrccI*/*xpf*, to *ku80*/*xrccI*/*xpf*/*xrcc2*) progressively decreased stable transformation from 2.6- to 5.4-fold, with little decrease in transient transformation. However, stable transformation was not eliminated, indicating that some other pathway(s) must exist for stable transformation to occur.

More recently, our laboratory has re-examined these same mutants (single, double, triple, and quadruple; [103]) for transient and stable transformation, for T-DNA integration under non-selective growth conditions, and for the growth and developmental phenotypes of these mutants [112]. As more mutations in DNA repair pathways were introduced, the plants became progressively more sensitive to light, and mutant plants grown under moderate light intensity displayed growth and developmental phenotypes. Using multiple serial dilutions of *Agrobacterium* for inoculation, major decreases in transient transformation, as well as stable transformation, were revealed in these mutants. However, T-DNA/plant DNA junctions isolated from single and *ku80*/*xrccI* double mutants had characteristics similar to those isolated from transformed wild-type Col-0 plants. Finally, using droplet digital PCR to evaluate *Agrobacterium*-infected calli grown under non-selective conditions, little difference existed in the amount of T-DNA integrated into the genomes of wild-type calli and calli derived from the various DNA repair mutants. Although these results differ quantitatively from those of Mestiri et al. [103], they confirm that simultaneous mutation of genes in these various DNA repair and recombination pathways do not eliminate either stable transformation or T-DNA integration. Therefore, there must be some other pathway available for integration to occur.

## 8. Where Do We Go from Here?

Studies on the importance of DNA repair genes and pathways in T-DNA integration have, for the most part, depended on a genetic approach to examine stable *Agrobacterium*-mediated transformation. The potential problems of using stable transformation as a proxy for T-DNA integration have been discussed above. However, a genetic approach may also have limitations: Genes important for T-DNA integration may be essential for cell viability, and homozygous mutation of these genes may be lethal to the organism. An example of a gene that may fit this category is that encoding DNA ligase I (*Lig1*). Because of its role in DNA replication, homozygous mutants in *lig1* are lethal [113]. Therefore, one may not expect to recover a homozygous *lig1* mutant when screening for T-DNA integration-deficient plants. It may be possible to screen partial loss-of-function *lig1* mutants for disruption of T-DNA integration. However, considering the important role Lig1 protein plays in fundamental cellular processes, interpretation of these integration data may be difficult.

A complementary approach to understanding proteins and pathways essential for T-DNA integration may be the use of biochemistry to dissect the integration process. Ultimately, development of an in vitro T-DNA integration assay, with fully characterized components, may be required to dissect the integration process. The beginning of such as assay was described previously [26]. However, this assay used a mostly uncharacterized plant cell lysate.

A major difficulty in studying T-DNA integration is that it is random, and therefore difficult to analyze biochemically in a large plant genome. Because double-strand DNA breaks tend to “trap” T-DNA, one approach to investigating the integration process may be to use CRISPR technology to generate a double-strand DNA break at a precise genomic location just prior to/concomitant with transformation. This will allow scientists to concentrate their biochemical methodologies on a specific locus. For example, one could analyze *Agrobacterium* and/or plant proteins, including those affiliated with T-strands, that are attracted to these breaks.

Another approach would be to use specific and more readily accessible molecular forms of T-DNA to investigate the roles of proteins (both known or surmised) in alternative “integration-like” mechanisms that accurately reflect T-DNA integration. To this end, the characterization of T-circles isolated from *Agrobacterium*-infected plants may prove fruitful. T-circles are double-strand circular molecules that contain T-DNA that is joined at or near T-DNA RB and LB sequences [114,115]. They are formed in the plant and have many characteristics of T-DNA/plant DNA junctions, although they are not integrated into the plant genome. For example, T-circles can consist of T-DNA dimers in direct (RB-to-LB) or inverted (RB-to-RB or LB-to-LB) orientation, similar to what is frequently seen with integrated T-DNAs. RB sequences can be precise (i.e., cut between nucleotides 3 and 4 of the 25 bp RB) or have small deletions. Deletions at the LB are more common and more extensive. Filler DNA can appear between the RB and LB sequences [115].

More recent experiments analyzing hundreds of T-circles confirmed and extended information about T-circle RB and LB junctions [116]. These additional features include identification of filler DNA between the RB and LB as coming from the plant genome or from other replicons in *Agrobacterium*, including the pAtC58 “cryptic” plasmid, the vir helper plasmid, or regions from the binary vector backbone. Filler DNA from these genomic sources was detected previously (Supplementary Table S1 of [106]). Some of these vector backbone sequences are in “inverted orientation” relative to the orientation of the border sequences, a feature found in integrated T-DNAs [44]. Microhomology frequently exists between this filler DNA and the adjacent T-DNA sequences. In addition, T-circles can be recovered from *Arabidopsis ku80* mutant plants at the same frequency that they are recovered from wild-type plants, and the RB–LB T-circle junctions are similar regardless of whether they are isolated from wild-type or *ku80* mutant plants. These characteristics are the same as those seen in integrated T-DNA/plant DNA junctions [104,112]. Thus, in all aspects examined to date, T-circle border junctions resemble those of integrated T-DNA. Because of these similarities, T-circles may serve as a more readily accessible proxy for studying the mechanism of, and the proteins important for, T-DNA integration.

## 9. Important Remaining Questions to Answer Regarding T-DNA Integration

Many important questions remain for investigating T-DNA integration and the role that plant DNA break repair processes may play. Among these are:Are plant single-strand nicks, double-strand breaks, neither, or both required for T-DNA integration?Are there particular aspects of plant chromatin that are more conducive for integration to occur?What are the roles, if any, of *Agrobacterium* virulence effector proteins (such as VirD2, VirE2, and possibly others) in T-DNA integration?What plant proteins are important for integration? Related to this question, what (if any) plant DNA repair pathway(s) is/are used for integration?Transient transformation of most plant species and tissues is considerably more efficient than is stable transformation. Why is this? If many T-strands enter the nucleus and can initially be converted to double-strand transcription-competent forms (either linear or circular), why is T-DNA integration relatively rare in these nuclei? Could greater expression of plant DNA repair genes important for T-DNA integration increase the percentage of stably transformed cells?What form of T-DNA (single-strand, double-strand linear, double-strand circular) is the substrate or template for integration?What is the molecular basis for differences in the frequency of T-DNA integration among plant species, or even among varieties/cultivars of the same species?During transformation, many plant cells are exposed to *Agrobacterium*, but only a few may be transformed (either transiently or stably). What is the basis for plant cell transformation competency?

## Figures and Tables

**Figure 1 ijms-22-08458-f001:**
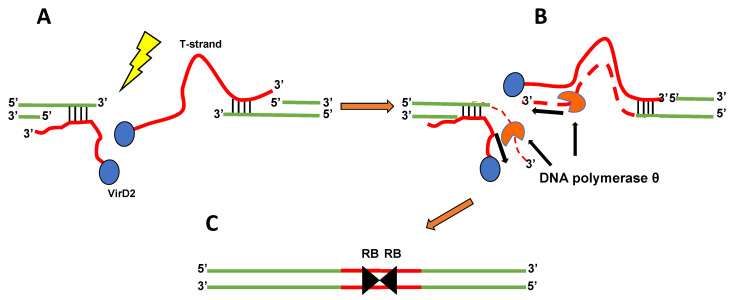
Model for the involvement of DNA polymerase θ in T-DNA integration. (**A**) Following generation of a double-strand DNA break (lightning bolt), the 5′ ends of plant DNA (green lines) are resected. Sequences within single-strand T-DNA molecules (T-strands, red lines), capped with VirD2 protein (blue oval) at their 5′ ends, align by microhomology with complementary regions of the exposed single-strand plant DNA. This can occur at both sides of the break (as pictured) or at one side only. (**B**) DNA polymerase θ (orange “pac-man” symbol) copies the T-strand sequence, using the exposed 3′ end of the plant DNA as a primer. The newly synthesized T-strand complementary sequence is ligated to the 5′ end of the break site, and the 5′ end of the T-strand is ligated to the 3′ end of the broken plant DNA following VirD2 removal. (**C**) Final result of integration of two T-DNA molecules in a head-to-head (RB-to-RB) orientation.

**Figure 2 ijms-22-08458-f002:**
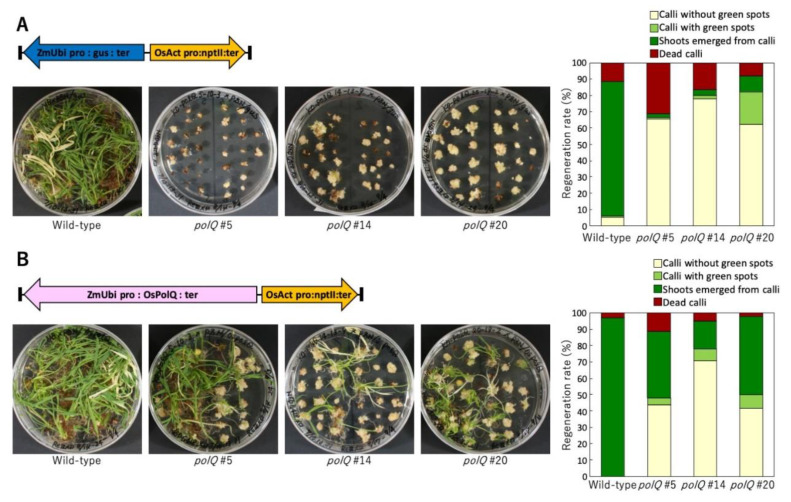
Regeneration of wild-type and *polQ* mutant rice plants after transformation by an *Agrobacterium* strain lacking or harboring a wild-type rice *PolQ* gene in the T-DNA. Transgenic calli (wild-type Nipponbare or three different *polQ* mutant lines (see [106]) were selected on medium containing 35 mg/L geneticin (G418) and 25 mg/L meropenem for four weeks. The calli were then transferred to regeneration medium (ReIII) lacking geneticin. Left panels, the plates were photographed after further incubation for four weeks. Right panels, the percentage of calli that developed shoots was quantified. (**A**) Infection by an *Agrobacterium* strain lacking a *PolQ* gene in the T-DNA. (**B**) Infection by an *Agrobacterium* strain containing a *PolQ* gene in the T-DNA. Data are from Dr. Ayako Nishizawa-Yokoi.

**Figure 3 ijms-22-08458-f003:**
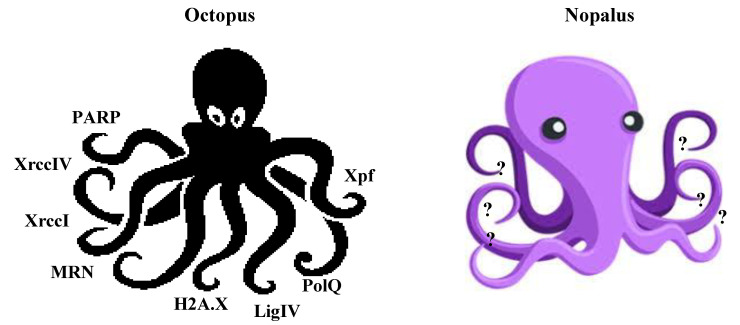
“Fantastic” models for mechanisms of T-DNA integration. Left, the octopus which grabs proteins from any NHEJ pathway it can to facilitate T-DNA integration. Right, the mythical nopalus, representing unknown pathway(s) for T-DNA integration. (The author thanks Barbara Hohn for suggesting the term “octopus” to describe one T-DNA integration model, and John Kemp for suggesting, in 1979, the term “nopalus” as a sarcastic alternative to “octopus”, from which octopine is derived, for the origin of nopaline. “Nopal” is a Spanish name for *Opuntia* cacti. Nopaline was first isolated from *Opuntia* crown gall tumors).

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
