# Peer review of "Plant DNA Repair and Agrobacterium T−DNA Integration"

_ijms, 2021, doi:10.3390/ijms22168458_

Round 1
Reviewer 1 Report
This Review presents deep and comprehensive analysis of Agrobacterium T-DNA integration mechanisms and involvement of plant DNA repair pathways into these processes. It represents a high quality manuscript with detailed description of bacterial and plant proteins involved into integration, integration hot spots, underlying mechanisms and future prospects. This Review is written by highly professional researcher in the field and will be useful for many specialists dealing with genetic engineering of plants and plant genetics, both from academic science and biotech companies. By my opinion it can be accepted without any revisions.
Author Response
Thank you for the excellent review.
Reviewer 2 Report
The manuscript “Plant DNA repair and Agrobacterium T-DNA integration” is a review on the process of T-DNA integration in plant genome. It is very interesting and well written, with a right balance between critic interpretation of the scientific literature, open questions, and future directions for research in this field.
I just think that it could be further improved by adding some considerations on how this piece of information can be taken into account when Agrobacterium-mediated transformation is used to study gene function. As an example, usually two independent lines are used to exclude the position effect, is that enough in light of this review?
Author Response
Thank you for the excellent review. As you suggested, I have added the following (lines 158-163):
"As described above, the position of T-DNA integration, and the chromatin structure of that region, may influence the expression of T-DNA-encoded transgenes. As a practical consideration, this variability in transgene expression (the so-called “position effect”) will influence studies on gene and promoter function. Scientists therefore need to examine a large number of independent transgenic events to draw conclusions about, e.g., relative promoter strengths."